# Hybrid denture acrylic composites with nanozirconia and electrospun polystyrene fibers

**A. A. Elmadani[1], I. Radović[2], N. Z. Tomić ⬤[3]\*, M. Petrović[1], D. B. Stojanović[1], R. Jančić Heinemann[1], V. Radojević[1]**

**1** University of Belgrade, Faculty of Technology and Metallurgy, Belgrade, Serbia, **2** University of Belgrade, Laboratory for Materials Sciences, Institute of Nuclear Sciences "Vinča", Belgrade, Serbia, **3** Innovation Center of Faculty of Technology and Metallurgy in Belgrade, Belgrade, Serbia

\* ntomic@tmf.bg.ac.rs

**Data Availability Statement:** All relevant data are within the manuscript.

**Funding:** The authors received no specific funding for this work.

## Abstract

The processing and characterization of hybrid PMMA resin composites with nano-zirconia ($ZrO_2$) and electrospun polystyrene (PS) polymer fibers were presented in this study. Reinforcement was selected with the intention to tune the physical and mechanical properties of the hybrid composite. Surface modification of inorganic particles was performed in order to improve the adhesion of reinforcement to the matrix. Fourier transform infrared spectroscopy (FTIR) provided successful modification of zirconia nanoparticles with 3-Methacryloxypropyltrimethoxysilane (MEMO) and bonding improvement between incompatible inorganic nanoparticles and PMMA matrix. Considerable deagglomeration of nanoparticles in the matrix occurred after the modification has been revealed by scanning electron microscopy (SEM). Microhardness increased with the concentration of modified nanoparticles, while the fibers were the modifier that lowers hardness and promotes toughness of hybrid composites. Impact test displayed increased absorbed energy after the PS electrospun fibers had been embedded. The optimized composition of the hybrid was determined and a good balance of thermal and mechanical properties was achieved.

## Introduction

Composite materials combine the properties of their constituents offering the new material improved properties and enabling the tuning of the properties to fit predefined needs. Hybrid reinforcement composite systems are created with the aim of improving t physical and mechanical properties by a synergy of two or even more reinforcement types. In the wide area of research, hybrid reinforcements were of different combinations: particles and fibers [1–4], two different types of particles [5], particles and whiskers [6], two or three types of fibers [7]. The improvement is, in general, better with multiple rather than a single reinforcement type, so that every one of the added reinforcements improves a different material property. One of the reinforcements should be aimed at improving toughness and the other, for example, improving hardness and elastic modulus [8].

**Competing interests:** The authors have declared that no competing interests exist.

The type, shape, and dispersion of fillers in a composite significantly influence the mechanical and thermal properties of the composite [9–13]. Reducing the size of particles from micro to nano size level may lead to an enhancement of the fillers' influence on the properties of the matrix. Among the various different classes of nanocomposites that have been developed over the past two decades, biocompatible nanocomposites have gained great attention from the research community due to their high potential in saving and prolonging human lives. Along with medicine, dentistry has been focused on the design of biocompatible materials that exhibit high mechanical endurance and chemical resistance with satisfactory aesthetic standards. Particularly, denture materials should express good impact resistance, hardness, and stability in the oral environment. Therefore, the focus in their design should be on the achievement of favorable mechanical properties and durability instead of reparation and replacement [14].

Acrylic resins are the most commonly used type of polymers for dental applications [15, 16]. They possess good chemical resistance, satisfy the aesthetic requirement and are easily processed. However, low impact resistance represents a serious drawback for the acrylic resins' use.

The idea of this work was to investigate the possibility of designing hybrid composite with specific properties—improved hardness on the surface and higher toughness in the center of the composite bulk [17, 18]. Nano-zirconia particles and electrospun non-wowen PS fibers were selected as reinforcements in the acrylic matrix. Ceramic reinforcements increase hardness and wear resistance of the composite, and the toughness can be improved with the addition of fine, continuous electrospun polymer fibers. The layered composite structure with altered layers of zirconia particles reinforced matrix and electrospun polymer fibers was prepared. Mechanical properties were tested in order to document the advantages of combining reinforcements as described.

Ceramic oxides, such as zirconia ($ZrO_2$), have been proven to be an excellent candidate for PMMA filler, due to its high hardness values and thermal resistance. Although ceramic materials can offer improvements in mechanical and thermal properties, their incompatibilities with polymers sometimes lead to agglomeration, diminishing the reinforcing potential of the nanoparticles. Improved ceramic/polymer adhesion can be achieved by coupling agents and nanoparticles coatings [19–23] Interface properties between nanoparticles and polymer matrix are modified by attaching the nanoparticles to the matrix. All these actions result in the improved mechanical and other functional properties of nanocomposites. Modification of nanoparticles can be chosen according to the polymer matrix by silanes, tetraethyl orthosilicate (TEOS), titanium isopropoxide (TIP), etc. [24–26]. For acryl-based matrices, silanes are the most used surface modifiers. Silanes are commonly tri-alkoxysilane esters, with three alkoxy groups directly bonded to the silicon atom [27–29].

Previous studies have also introduced dental composite materials with ceramic nanoparticles only, short fibers (glass, aramid, nylon, polypropylene, polyethylene, and others), electrospun polymer and ceramic nanofibers or electrospun nanofibers doped with nanoparticles [3, 30–36]. However, separate incorporation of nanofibers and nanoparticles could enable the modification of different properties by varying the constituent concentrations.

This paper is aimed at optimizing the composition of a hybrid acrylic composite with $ZrO_2$ nanoparticles and electrospun polystyrene (PS) fibers and to explore their influence on thermal and mechanical properties of the obtained nanocomposites. Zirconia nanoparticles were functionalized with 3-Methacryloxypropyltrimethoxysilane (MEMO) silane to improve matrix-particle bonding. It is well known that zirconia nanoparticles enhances the hardness, thermal and wear properties, while polymer fibers improve toughness [37]. These mechanical properties are very important for polymer composites in exploitation, especially in dentistry.

Although the static mechanical behavior of composites in dentistry is well known and widely investigated, the dynamical loading and conditions under fracture are still developing [38, 39]. This work will aid researchers in dealing with the optimization of processing parameters for the production of composite materials with the desired advanced properties.

## Experimental

### Materials

Commercial acryl denture material „Simgal-Acryl R", Galenika AD, Belgrade, Serbia, was used as a polymer matrix. It is a two-component system consisting of a powder and a liquid. The powder consists of a PMMA copolymer and the initiator benzoyl peroxide (BPO) in a concentration of 1.1% w/w. The liquid was made of methyl methacrylate (calc) 94.15% w/w; acid as methacrylic acid 19.8 ppm w/w; $N$, $N$-dimethyl-$p$-toluidine as accelerator 0.85% w/w; ethylene glycol dimethacrylate as cross-linking agent 5.00% w/w; water 27 ppm w/w. Nanopowder of $ZrO_2$ (with a particle size ~100 nm), Sigma Aldrich, was used as particle reinforcement in the composite. 3-Methacryloxypropyltrimethoxysilane (MEMO) (Dynasylane, Evonik Industries) was used for surface modification of zirconia. Toluene and hexane (Sigma Aldrich) were used as solvents. Polystyrene (PS) used to obtain electrospun fibers was purchased as Empera®251N from Ineos Nova. Solvent for PS solution was 99.8% dimethylformamide (DMF), purchased from Sigma-Aldrich.

### Modification of zirconium oxide nanoparticles

5 g of $ZrO_2$ nanoparticles were dispersed in 150 ml of toluene in a round-bottom flask equipped with a reflux condenser under the flow of nitrogen. When the boiling point of toluene was reached, 1 g of MEMO silane was added and the resulting white suspension was stirred and refluxed for 22 h. After the completion of the reaction, the particles were filtrated and washed with hexane to remove the unreacted silane. The particles were dried at 40 ˚C in an oven for 12 h and then used for the preparation of nanocomposites [40].

### Electrospinning of PS fibers

Electrospinning (Electrospinner CH-01, Linari Engineering) was performed with a 20 ml plastic syringe with a metallic needle of 1 mm inner diameter set vertically on the syringe pump (R-100E, RAZEL Scientific Instruments) with 15 cm distance from the needle tip to the collector, and the high-voltage power supply (Spellman High Voltage Electronics Corporation, Model: PCM50P120) set to a voltage of 28 kV at the room temperature (25˚C) and the humidity of 47%. The flow rate of the polymer solution was 5.0 ml/h. The concentration of PS in DMF solution prepared for electrospinning was 22 wt. %.

### Composite preparation

A neat polymer matrix was obtained by mixing a two-component system (liquid volumetric ratio of 2.5:1) for 30 seconds. After that, the paste was processed in an aluminum mold under mechanical pressure with the room temperature of polymerization for 20 minutes. All samples had the dimensions required for the impact test (60 x 60 x 3.5 mm). For the composite processing, the particles were surface-modified to obtain a good dispersion. The nanoparticles were first dispersed in a liquid monomer in an ultrasonic bath for 1 hour and then mixed with a powder to initiate the polymerization. After that, the paste was poured in a mold. The samples with PS fibers were produced (in the mold) by a modified lay-up process; alternating layers of

initiated paste with nano zirconia and electrospun fibers. The compositions of a series of samples that were prepared are presented in **Table 1**.

## Characterization

The microstructure of the composites was studied by SEM microscopy using a Tescan Mira3 XMU field emission scanning electron microscope (FE-SEM) operating at 10 kV. A thin gold layer was deposited on the specimen surfaces before examination.

Image analysis was performed by Image Pro-Plus 4.0 software (Media Cybernetics) that provided the information about PS fiber diameter distribution.

Fourier transformed infrared (FTIR) analysis was performed to investigate bonding between $ZrO_2$ nanoparticles and the matrix. FTIR spectra of the samples in KBr discs were obtained by transmission spectroscopy (Hartmann & Braun, MB-series). The FTIR spectra were recorded between 4000 and 400 $cm^{-1}$ wavenumber region at a resolution of 4 $cm^{-1}$.

Thermal analysis of composites was performed on a device for differential scanning calorimetry (DSC) in a temperature range from 24°C to 160°C (Q10, TA Instruments) under a dynamic nitrogen flow of 50 ml min–1. Samples of 7–9 mg were investigated. The samples were heated up at a rate of 10°C min–1. The glass transition temperature was determined at the midpoint of the step-transition for each sample. The $Tg$ values were confirmed by the use of the derivative curve.

Mechanical characterizations of the samples were performed by Vickers microhardness (HV) tester "Leitz, Kleinharteprufer DURIMET I", using a load of 4.9 N. The loading time was 15s. Six indentations were made, yielding twelve indentation diagonal measurements, from which the average hardness could be calculated. The indentation was performed at room temperature.

Impact test was performed using Puncture Impact testing machine HYDROSHOT HITS-P10. The clamping plates with an aperture 40 mm in diameter and clamping pressure of 0.55 MPa were used. The striker with a hemispherical head, 12.7 mm in diameter, was loaded with programmable velocity, height and attained value of depth. In that manner, it was possible to control the impact energy. The data for the force, deflection, velocity and energy with time were recorded. The impact speed was set at 1 m/s and the maximum load was 10 kN. This loading regime could be considered as intermediate, which is perfectly appropriate for denture loading conditions [39, 40]. All the samples were of the same dimensions (60 x 60 x 3.5 mm). Tests were performed on five specimens according to the ASTM D 3763–15, and the results were presented as mean values with standard deviations. The data were analyzed in terms of the maximum load, energy corresponding to the maximum load and total energy.

**Table 1. Samples used for the comparison of thermal and mechanical properties.**

| Sample | $ZrO_2$, wt. % | PS fibers, wt. % | $ZrO_2$/MEMO, wt. % |
|---|---|---|---|
| A (pure PMMA) | 0 | 0 | 0 |
| A-$ZrO_2$ | 1.0 | 0 | 0 |
| A- $ZrO_2$/MEMO | 0 | 0 | 1.0 |
| A-PS | 0 | 2.5 | 0 |
| A-PS-$ZrO_2$ | 1 | 2.5 | 0 |
| A-PS-$ZrO_2$/MEMO-0.5 | 0 | 2.5 | 0.5 |
| A-PS-$ZrO_2$/MEMO-1.0 | 0 | 2.5 | 1.0 |

## Results and discussion

### FTIR analysis

The FTIR spectra of both unmodified and modified zirconia nanoparticles and the composite with modified zirconia are presented in **Fig 1A**. All spectra have a peak at 754 cm−1 that is attributed to Zr–O stretching vibrations at $ZrO_2$ nanoparticles. Characteristic acrylate $CH_3$ vibration of MEMO silane was observed at 1173 cm$^{-1}$ in the spectrum of modified zirconia. The peak at 1721 cm$^{-1}$ that is associated with carbonyl stretching band C = O which is present in the silane coupling agent (MEMO) was observed in the spectrums of modified zirconia and the composite (shifted to 1733 cm-1) [41–43]. The presence of adsorbed water was confirmed by the Zr–$H_2O$ flexion at 1635 cm$^{-1}$.

FTIR spectrums of A-PS-$ZrO_2$ and A-PS-$ZrO_2$/MEMO are presented in **Fig 1B**. All the spectra have peaks in the region of 2995–2840 cm$^{-1}$, which are assigned to the stretching of the C–H bonds contributed mostly to PMMA and PS. The peak at 749 cm$^{-1}$ is attributed to Zr–O stretching vibrations from nanoparticles in all the spectrums. Double bond C = C stretch which is sensitive to ring strain vibration at 1649 cm$^{-1}$, indicated conjunction with the phenyl group in PS [44– 46], emphasized with MEMO silane. An increased intensity of the signal at 960 cm$^{-1}$ in A-PS-$ZrO_2$/MEMO compared to A-PS-$ZrO_2$ indicates the formation of Si–O–Zr bond [45].

### FE-SEM analysis

Morphology and size of PS fibers were observed by FE-SEM analysis. The size distribution of PS fibers was obtained using image analysis tools and the results are presented in **Fig 2**. PS diameter distribution with mean diameter $D_{mean}$ = 1.51 μm (standard deviation = 0.52 μm) was best fitted with the Lognormal distribution curve.

FE-SEM images of cross-sections of the polymer after the impact testing are presented in Fig 3. Shows that zirconia agglomerates observed in the sample with unmodified particles had larger diameters and consisted of a larger number of individual particles, (**Fig 3A**) while surface modification of nanoparticles with MEMO silane (**Fig 3B**) enabled aggregates to be smaller in diameter and more evenly spaced. In **Fig 3C** and **3D** the areas with fibers are presented. The modification of nano zirconia with MEMO silane produced a monolayer of silane on the surface of the particles, and promotes deagglomeration in the polymer matrix because of the steric hindrance [47].

### DSC analysis

Results of DSC analysis are presented in **Fig 4** and **Table 2** with corresponding values of glass transition temperature (*Tg*). Because there is no miscibility of PMMA and PS, there is no evident change in *Tg* values [48]. Also, it is possible that the values of *Tg* for PS and PMMA could have overlapped each other with additional curing. Zirconia behave as highly functional physical cross-links, and hence reduce the overall mobility of the polymer chains, even when interactions with the polymers are only on a physical level [49, 50]. The embedding of modified nano zirconia slightly increases the Tg of the composite as a consequence of an interaction between the modified zirconia interface and acrylic resin [51]. Interfacial Si-O bond formation on the surface of zirconia enables chemical bonding with polymer matrix [50–54]. This also leads to better deagglomeration of nanoparticles. In this case, the mobility of polymer chains was suppressed even better, and *Tg* for this composite is the highest (Table 2). This further indicates that the thermal properties of the hybrid can also be adjusted with the optimal ratio of zirconia and PS fibers in the composite.

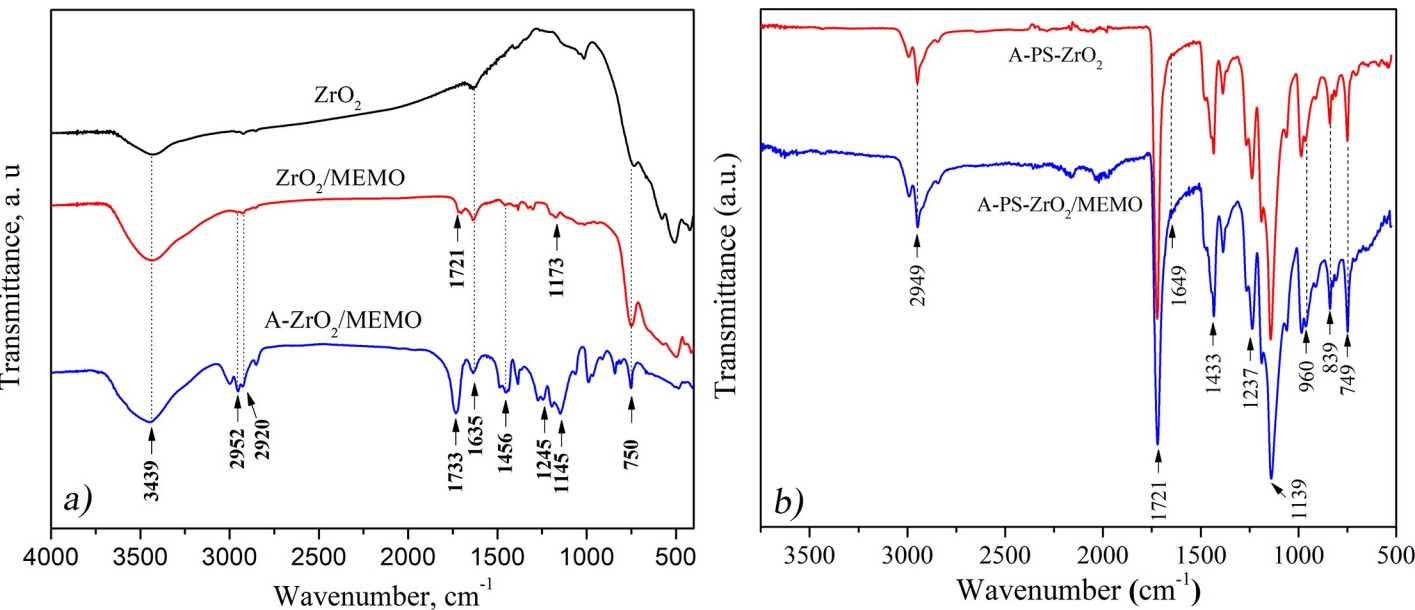

**Fig 1.** FTIR spectra of: a) neat ZrO₂ particles, ZrO2 particles modified with MEMO silane and a composite reinforced with ZrO₂/MEMO and b) hybrid composites A-PS-ZrO₂ and A-PS-ZrO₂/MEMO.

### Vickers hardness test

Vickers hardness test reflects the uniformity of reinforcement dispersion in the composites and its resistance to shear stresses under local volume compression. **Table 3** presents Vickers values for the PMMA matrix and the composites. The addition of 1 wt. % zirconia nanoparticles improved microhardness by 3%. In composites with silanized zirconia (ZrO₂/MEMO) the effective dispersion and cross-linking was achieved, and thus improvement of HV value of

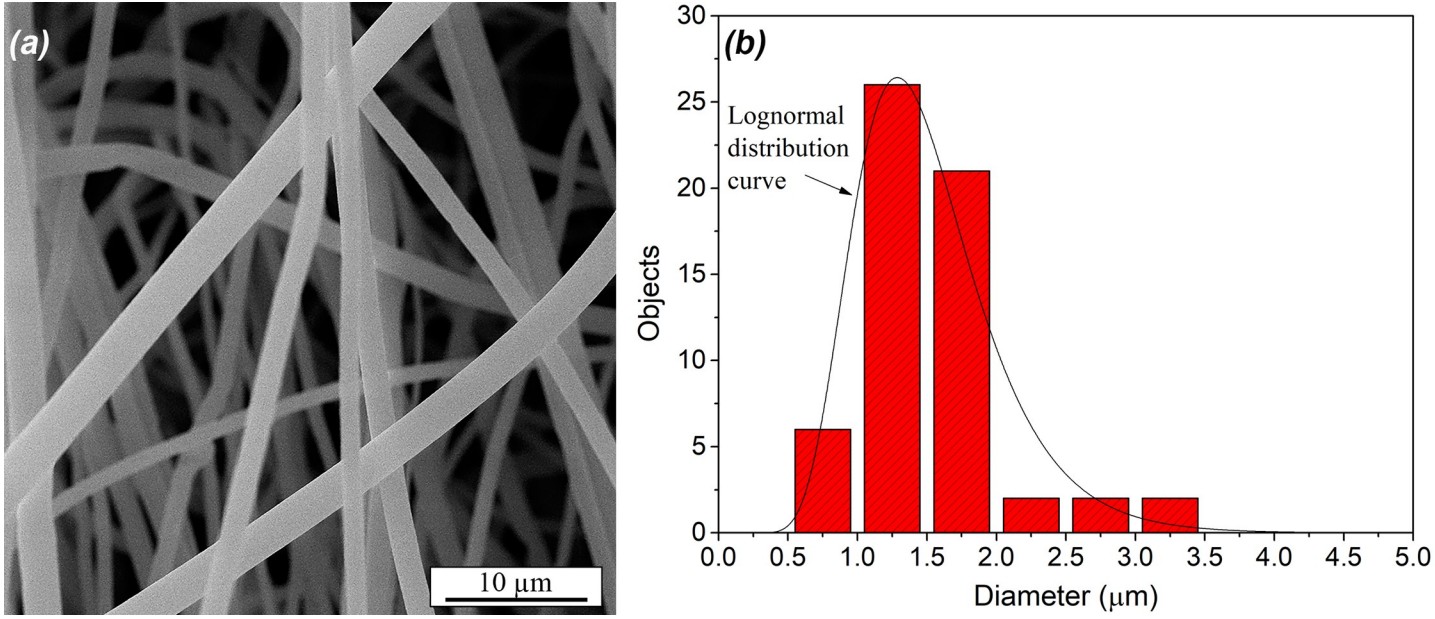

**Fig 2.** (a) FE-SEM micrograph of PS fibers and (b) size distribution determined by image analysis.

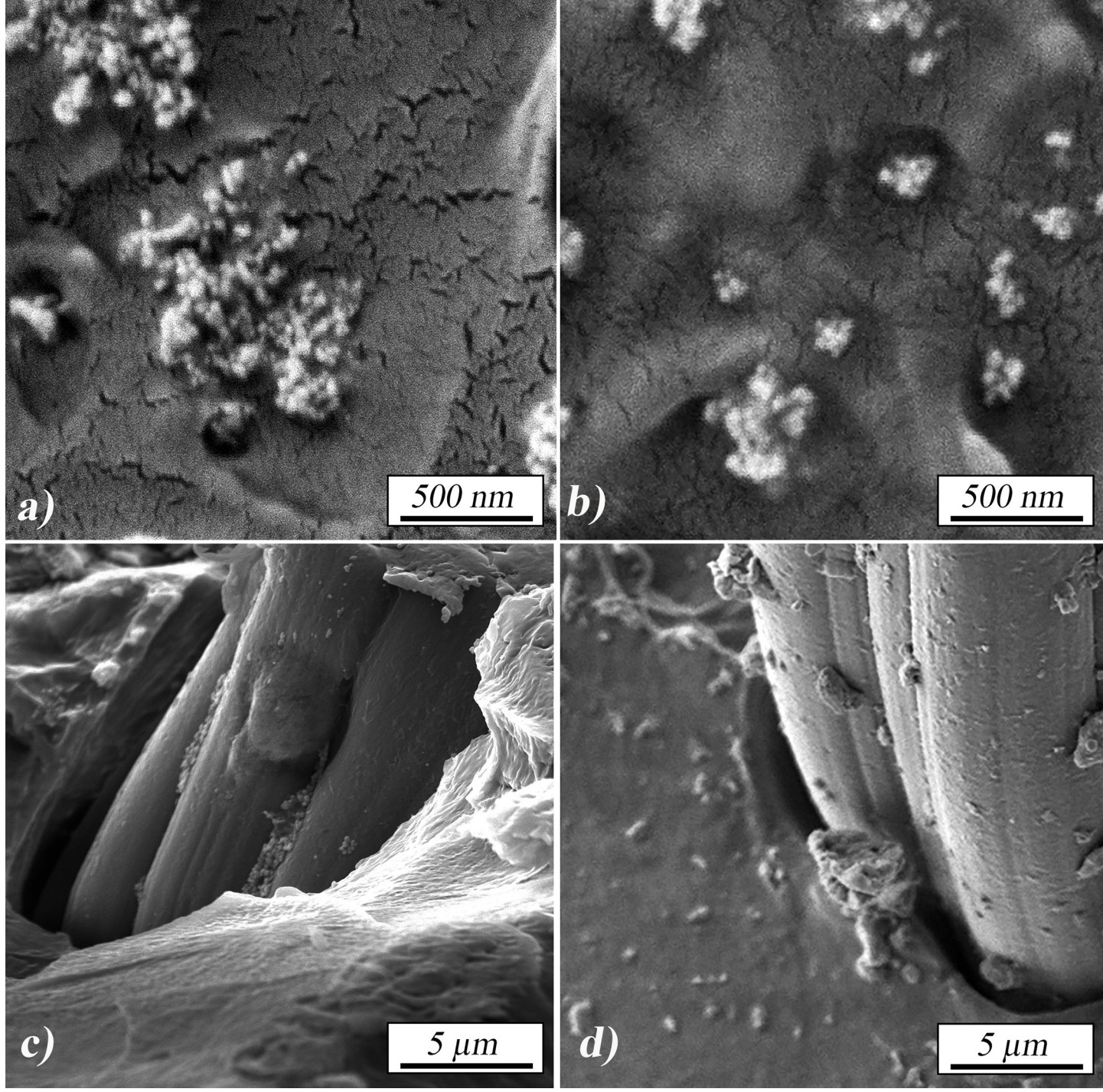

**Fig 3.** FE-SEM impact fracture surface images of: a) and c) A-PS-ZrO$_2$; b) and d)) A-PS-1% ZrO$_2$/MEMO.

29% for 1 wt. % modified zirconia resulted. On the other hand, the introduction of PS fibers leads to lower hardness values. The PMMA and PS are immiscible and don't interact easily. Added PS fibers didn't interact with the PMMA paste during the preparation of the composite.

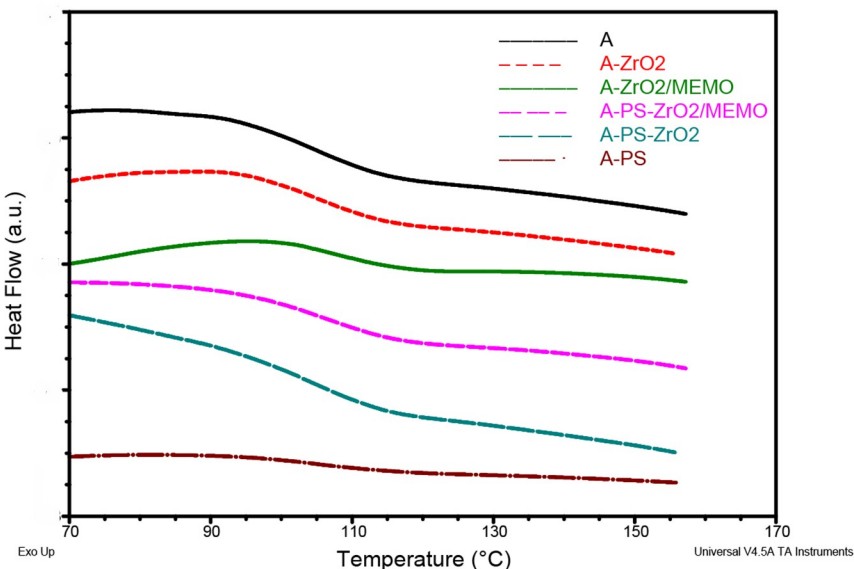

**Fig 4. DSC curves of polymer matrix and composites.**

But presence of PS fibers have influenced polymerization of PMMA, and resulted of phase separation in hybrid composite.

This leads to interfacial tension in those areas which is followed by attraction of nanoparticles in the vicinity of PS fibers. This influence of the composites' mechanical behavior of leads to lower hardness and *Tg* [55–57].

In order to emphasize the influence of different nanoparticle concentrations in the hybrid composites, samples with 0.5 wt. % of MEMO silane functionalized $ZrO_2$ nanoparticles were also subjected to micro Vickers test. The hardness growth trend remained the same–nanoparticles $ZrO_2$/MEMO offer improved hardness due to higher compatibility of MEMO functional groups with PMMA matrix. The presented results revealed that the hardness could be adjusted by optimizing the content ratio of modified zirconia and PS fibers.

Microhardness testing is based on the local plastic deformation of a sample under applied stress in the vicinity of the indenter. Glass transition temperature depends of the structure and morphology of polymer chains and its behavior in higher temperatures. In glassy state critical stress for plastic deformation of amorphous polymer requires movements of macromolecule bundles against the resistance of the stiff chain segments., hardness for amorphous polymers could be higher than for semi-crystalline polymers at temperatures under *T*g.

It is well known that polymer composites in the glassy state are sensitive to free volume change, and the *T*g and hardness could correlate to this [58–62]. The temperature coefficients

**Table 2. DSC results for all PMMA samples.**

| Sample | $T_g$, ˚C |
|---|---|
| A | 104.1 |
| A-$ZrO_2$ | 104.5 |
| A- $ZrO_2$/MEMO | 108.0 |
| A-PS | 103.3 |
| A-PS-$ZrO_2$-1.0 | 103.9 |
| A-PS-$ZrO_2$/MEMO-1.0 | 104.5 |

**Table 3. Results of Vickers hardness test.**

| Sample | HV, MPa | St.dev., MPa |
|---|---|---|
| A | 243 | ±1 |
| A-ZrO$_2$ | 250 | ±5 |
| A- ZrO$_2$/MEMO | 313 | ±8 |
| A-PS | 211 | ±5 |
| A-PS-ZrO$_2$-1.0 | 229 | ±8 |
| A-PS-ZrO$_2$/MEMO-0.5 | 232 | ±1 |
| A-PS-ZrO$_2$/MEMO-1.0 | 269 | ±3 |

of the molar volume, free volume and enthalpy change of the glass–rubber transition are closely related to the cohesive energy density of the polymer. The CED is also the main factor determining hardness. Many analytical theories correlate with this phenomenology applied to nanocomposites [62–67]. This correlation of the composites examined in this work is presented in **Fig 5**. There was a good linearity obtained with a correlation coefficient $R^2$ = 0.93321. This result is in agreement with the assumptions of the influence of polymer matrix morphology on the mechanical properties of composites [58]. In an amorphous polymer matrix, the filler can be distributed freely. Composite with surface-modified nanoparticles provides stronger resistance to plastic deformation as it is chemically bound to the matrix. The addition of polymer fibers leads to certain relaxation in mechanical response to indentation, while in the hybrid composite some of the nano zirconia were constrained between PS fibers.

## Impact test

The position of a sample in the impact machine and samples before and after the impact test is presented in **Fig 6**

The results of the controlled energy impact test are presented in **Fig 7**. The impact behavior of the hybrid nanocomposites with PS fibers and modified particles was significantly improved, compared to the pure PMMA. **Table 4** presents the absorbed energy values. Absorbed energy is defined as the energy difference between the total energy and the energy at peak load. As the composite materials are brittle, it was assumed that energy up to the peak load was due to the elastic deformation of the sample and that beyond the peak load, energy was spent on creation and propagation of cracks.

As expected, sample A-PS showed the highest ability to absorb energy during the impact, almost 92% higher than the pure PMMA.

FE-SEM analysis showed the difference in the observed matrix-PS fiber interface after the impact test. Acrylic resin with PS fibers has a clear and smooth surface that indicates poor contact between the two polymers (**Fig 8A**). In the case of a matrix with unmodified ZrO2 particles, it could be seen that particles built some agglomerates (**Fig 8B**) at PS surface and slightly improved contact between the fibers and the matrix. Surface modification of ZrO$_2$ particles with MEMO silane significantly improved the compatibility of the interfaces in a hybrid composite A-PS-ZrO$_2$/MEMO (**Fig 8C**). Modified composite matrix (A-ZrO$_2$/MEMO) filled the space between the fibers and improved the contact between the matrix and the fibers. This indicated that the interactions between the fibers and the matrix were strong enough to allow load transfer from the matrix to the fibers, which should ensure better mechanical properties of the processed hybrid nanocomposite [41, 42]. The interfacial adhesion strength between the matrix and the fibers affected the impact property of composites and resistance to crack generation and propagation [65–69]. These failure mechanisms led to the high absorption of impact

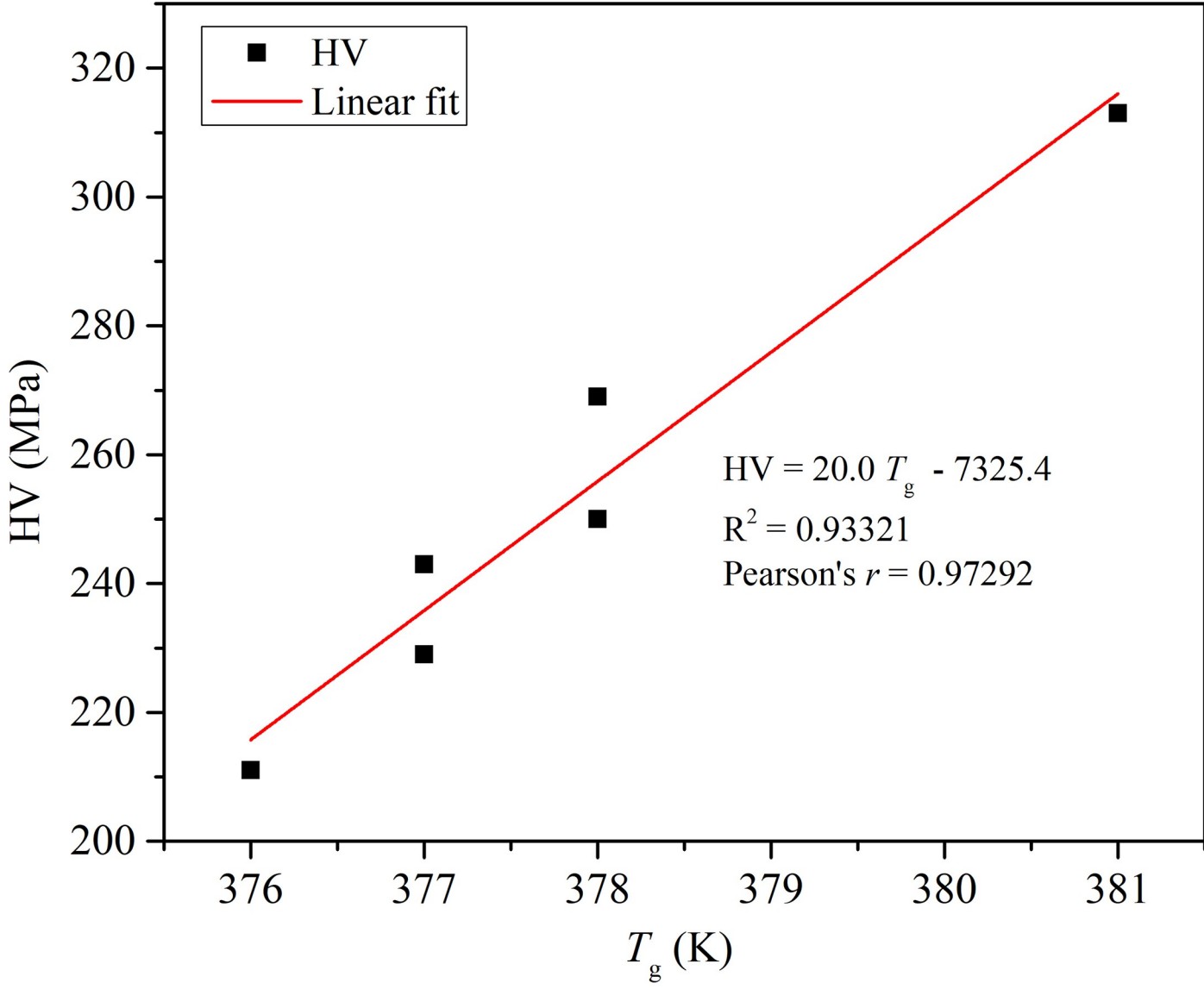

**Fig 5. Linear correlation of $HV$ and $T_g$ obtained from DSC and microhardness analysis.**

energy as a consequence of dissipation during crack propagation. FE-SEM analysis **Fig 8A and 8B**) indicates a brittle fracture tendency in specimens with PS fibers and hybrids with untreated zirconia (river pattern associated with crack propagation).

For the hybrid composites with modified zirconia, **Fig 8C**, with the presence of flexible – O–Si–O– bond [24–26] consequently provide better adhesion to the matrix and cracks found difficult to propagate. Brittle to ductile transformation of failure mechanism was obtained [65–67].

Results of impact behavior show the possibility of composite mechanical properties modulation, **Fig 8D** [70]. Incorporation of pure brittle zirconia, prone to agglomeration, reduced absorbed energy, while it was successfully compensated latter with the combination of modified particles and the electrospun polymer fibers. Therefore, sample A-PS-ZrO$_2$/MEMO-1.0

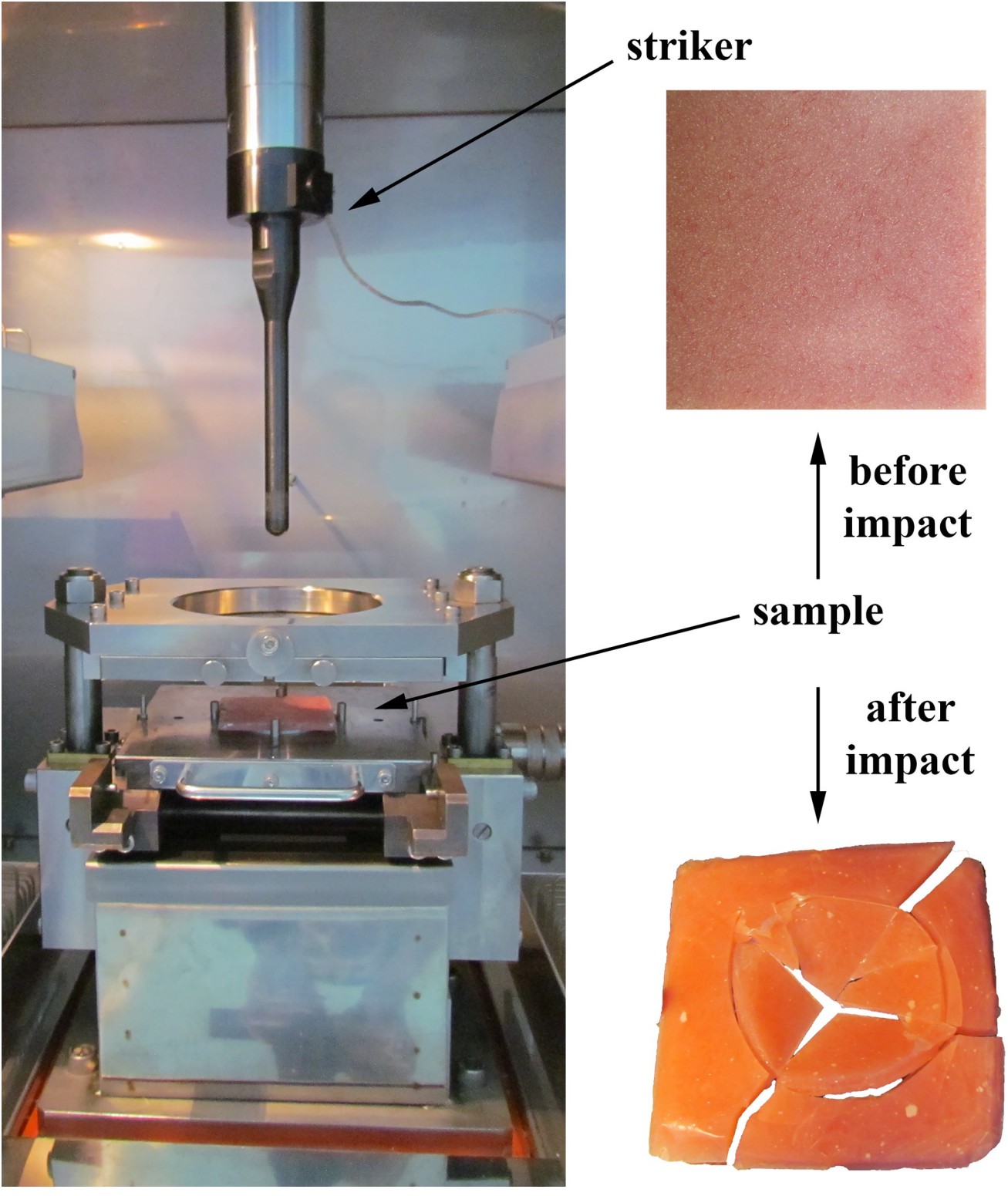

**Fig 6. Sample in the impact test machine before and after the impact test.**

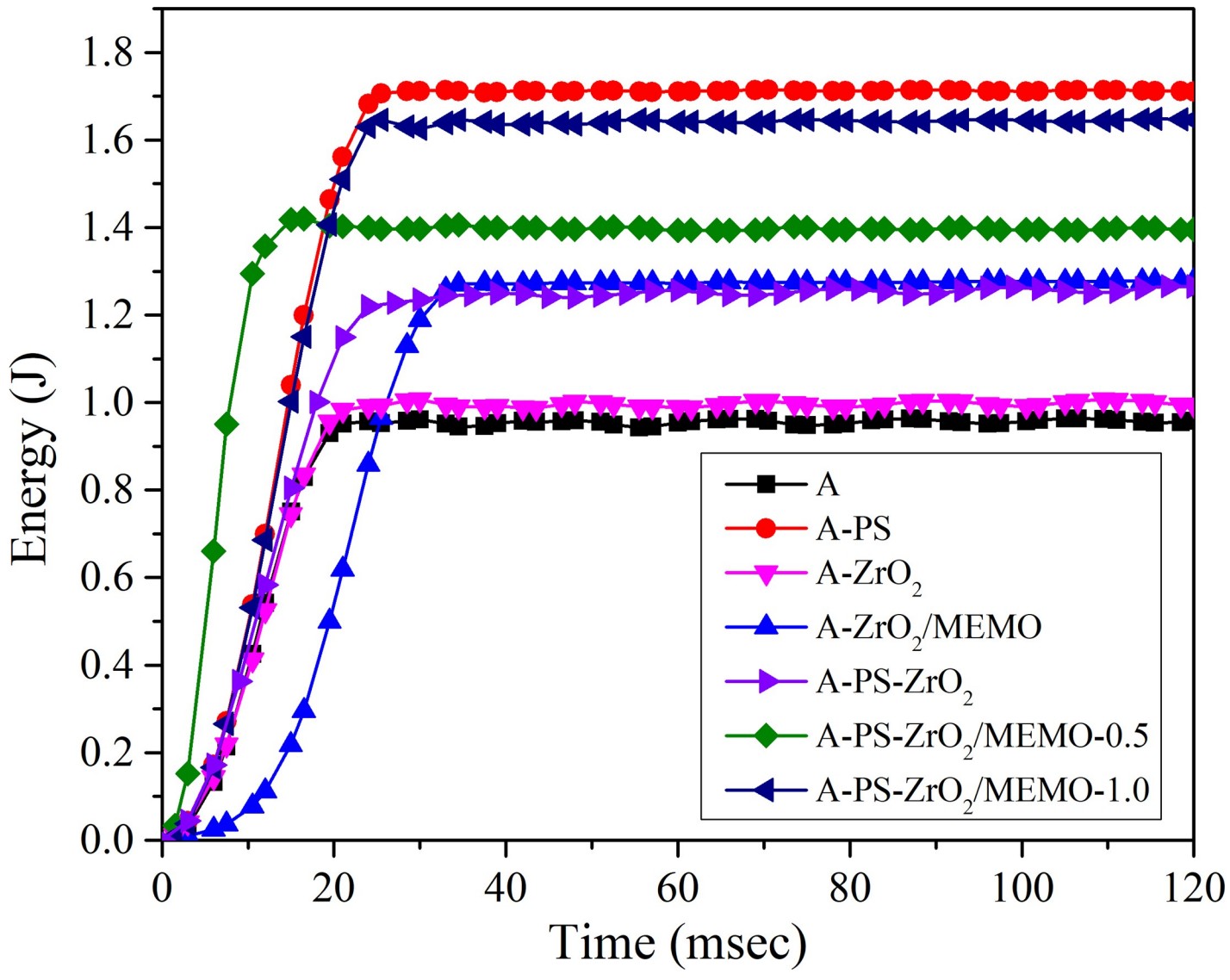

**Fig 7. Energy-time curves obtained from impact test.**

**Table 4. Absorbed energy during impact.**

| Sample | $E_{abs}$, J | SD, J |
| --- | --- | --- |
| A | 0.46 | ±0.01 |
| A-PS | 0.84 | ±0.03 |
| A-ZrO$_2$ | 0.28 | ±0.01 |
| A-ZrO$_2$/MEMO | 0.55 | ±0.02 |
| A-PS-ZrO$_2$ | 0.51 | ±0.01 |
| A-PS-ZrO$_2$/MEMO-0.5 | 0.74 | ±0.04 |
| A-PS-ZrO$_2$/MEMO-1.0 | 0.79 | ±0.05 |

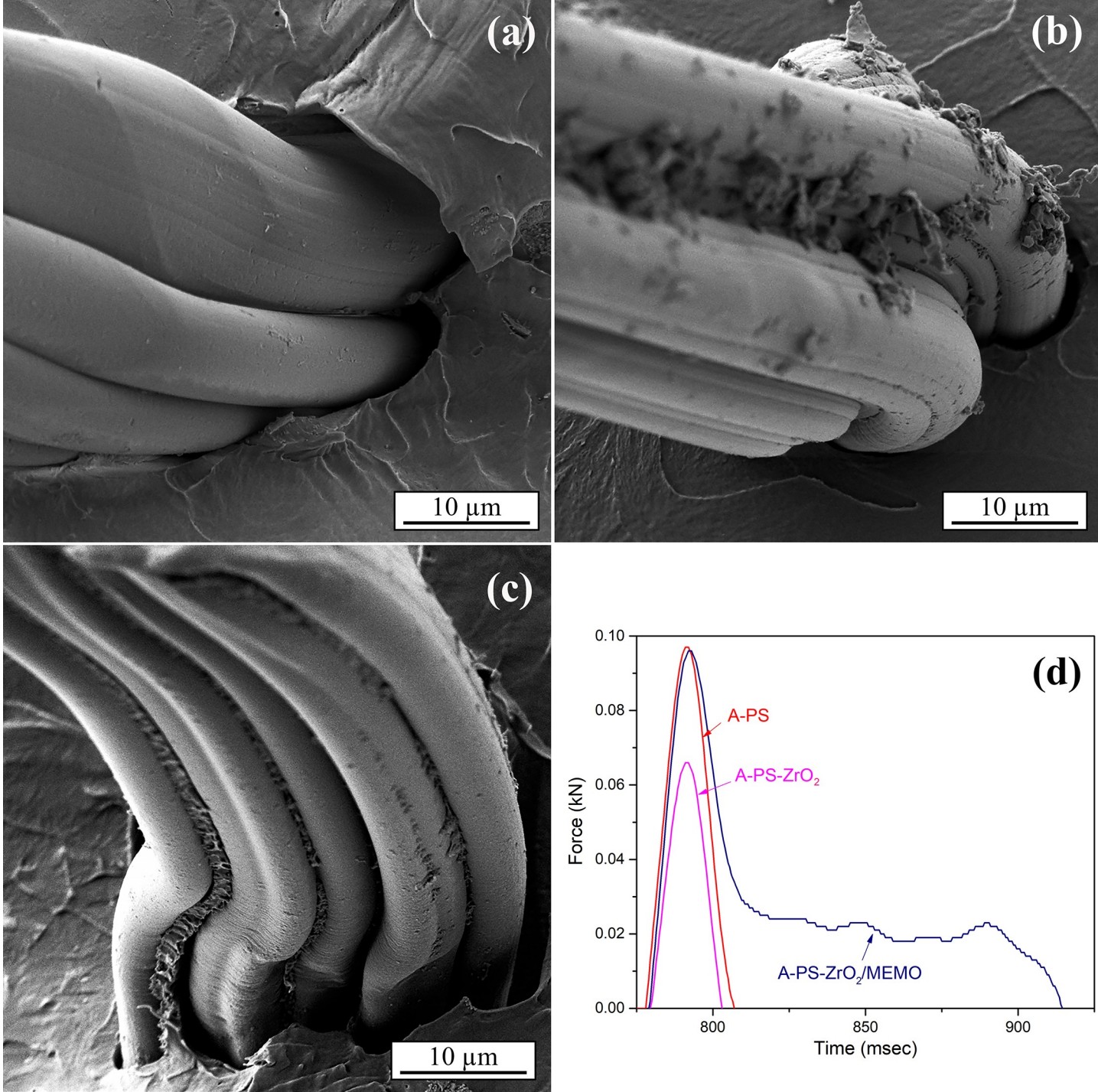

**Fig 8.** FE-SEM images of fracture surfaces: a) A-PS, b) A-PS-ZrO$_2$, c) A-PS-ZrO$_2$/MEMO and d) corresponding Force-time curves from impact test.

shows the value of total absorbed energy close to the one of A-PS, and the shape of failure mode which is, following the assumption, based on FE-SEM analysis.

**Table 5. Performance parameters comparison of the given system with the literature data.**

| | Matrix | Reinforcements | Mechanical properties | Reference |
|---|---|---|---|---|
| 1. | Acrylic Denture Base Material | 1% Silanized zirconium oxide (ZrO$_2$) nano filler and 2.5% electrospun PS fibers | Absorbed Impact energy improvement 70%, Hardness improvement 10% | This paper |
| 2. | Acrylic Denture Base Material | Silanized zirconium oxide (ZrO$_2$) nano filler and plasma treated polypropelene (PP) fibers | Tensile strength, improvement 44% | [1] |
| 3. | PMMA | PP fibers 2.5 t.%)/Al$_2$O$_3$ nanoparticles (1 wt.%) | Impact strength improvement 119% Surface hardness improvement 4.2% | [2] |
| 4. | Denture base PMMA | ZrO$_2$-Al$_2$O$_3$ | Tensile strength decrease 17.0% (highest value for 100% ZrO$_2$) Fracture toughness improvement 32.5% (ZrO$_2$/Al$_2$O$_3$ = 20/80) | [5] |
| 5. | Denture base PMMA | ZrO$_2$/Aluminum borate whiskers | Surface hardness improvement 26.4% (reinforcement ratio 1:2, 3% ZrO$_2$) Flexural strength improvement 52.3% (reinforcement ratio 1:2, 2% ZrO$_2$) | [6] |
| 6. | Denture base PMMA | Glass/ UHMW PE fibers in form of fabrics 5.3% | Flexural strength 90% and modulus 76% | [7] |

A comparison of results presented in this paper and the literature is presented in Table 5. It is evident that for some systems of hybrid reinforcements design of mechanical properties is possible with the proper combination of constituents according to exploitation requirements.

## Conclusion

Properties of novel hybrid resin composites with nano-zirconia and electrospun PS polymer fibers were presented in this study. The surface modification of ceramic reinforcement was introduced to improve the interaction of particles and the matrix. FTIR analysis confirmed the successful modification of nanoparticles with coupling agents, which provided better matrix-particle bonding and resulted in better mechanical properties. Furthermore, the silane modification of zirconia ensured more favorable dispersion, which further improved the stiffness of the composites. The research was carried out in order to determine the change in microhardness and impact behavior with different content of the constituents and their content ratio. Electrospun non-woven PS fibers should be a promising solution for the compensation of increased brittleness brought on by the incorporation of ceramic nanoparticles (that are) prone to agglomeration in the polymer matrix. DSC analysis was useful in the determination of the obtained composites' thermal properties. The slight change in $T$g was detected as an influence of different hybrid compositions. A nearly linear correlation between $T$g and microhardness was obtained, which could be explained by a change in cohesive energy density. Microhardness and impact test revealed that the optimal results were achieved by a combination of PS fibers and ZrO$_2$/MEMO in an acrylate matrix, where the fibers were able to compensate for brittleness caused by the ceramic nanoparticles. Moreover, the introduction of modified particles as reinforcement in the matrix improved the contact of the matrix to PS fibers and realized the optimal combination of materials' reinforcement in this composite. Ceramic nanoparticles, which possess appropriate surface modification; and non-woven polymer fibers are proven to be good candidates for incorporation in the acrylic matrix to successfully modulate of mechanical and thermal properties of the hybrid nanocomposite material.

## Author Contributions

**Data curation:** N. Z. Tomić.

**Formal analysis:** M. Petrović.

**Investigation:** A. A. Elmadani.

**Methodology:** I. Radović.

**Supervision:** V. Radojević.

**Validation:** D. B. Stojanović.

**Visualization:** R. Jančić Heinemann.

**Writing – original draft:** A. A. Elmadani.

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
