## [Decision Letter · Decision Letter 0]

16 Sep 2019

PONE-D-19-20656

Hybrid denture acrylic composites with nanozirconia and electrospun polystyrene fibers

PLOS ONE

Dear Dr Tomic,

Thank you for submitting your manuscript to PLOS ONE. After careful consideration, we feel that it has merit but does not fully meet PLOS ONE’s publication criteria as it currently stands. Therefore, we invite you to submit a revised version of the manuscript that CAREFULLY addresses the points raised during the review process.

We would appreciate receiving your revised manuscript by Oct 31 2019 11:59PM. To enhance the reproducibility of your results, we recommend that if applicable you deposit your laboratory protocols in protocols.io, where a protocol can be assigned its own identifier (DOI) such that it can be cited independently in the future. For instructions see: http://journals.plos.org/plosone/s/submission-guidelines#loc-laboratory-protocols

We look forward to receiving your revised manuscript.

Kind regards,

Yogendra Kumar Mishra, Ph. D.

Academic Editor

PLOS ONE

Journal Requirements:

"This work was supported by the Ministry of Science and Technological Development of the Republic of Serbia, Projects No. TR 34011 and III 45019"

Reviewers' comments:

Reviewer's Responses to Questions

**Comments to the Author**

1. Is the manuscript technically sound, and do the data support the conclusions?

Reviewer #1: Yes

Reviewer #2: Yes

2. Has the statistical analysis been performed appropriately and rigorously? 

Reviewer #1: Yes

Reviewer #2: N/A

3. Have the authors made all data underlying the findings in their manuscript fully available?

Reviewer #1: Yes

Reviewer #2: Yes

4. Is the manuscript presented in an intelligible fashion and written in standard English?

Reviewer #1: No

Reviewer #2: Yes

5. Review Comments to the Author

Reviewer #1: This manuscript reports the preparation of a hybrid nano-composites composed of nano-zirconia and electrospum polystyrene.

Results of the studies are very interesting an

1) first three sentences of the abstract are controversial and do not present manuscript well.

2) over all English needs careful proof reading

3) uniqueness aspects of the proposed work is not discussed well in introduction.

4) compare the performance parameters of this system with literature.

5) what are the prospects of this research.

Reviewer #2: The research work presented in the manuscript is scientifically sound and well written.

However, before accepting the same please do the following corrections:

1. Commercial acryl denture material „Akril R“, do the needful correction.

2. Fe-SEM images are not very clear and donot satisfy the claimed findings. Kindly elaborate further. Also, for comparison with the effect of PS, the image 3c should also be of 500 nm scale.

3. No rationale for better dispersion of ZrO2 with surface modification and presence of PS fibre is given. Explanation required.

4. Are the DSC images plotted (not original). Provide the original DSC images. Further, how much weight of each sample was taken for the analysis. The variation in the thermal stability and changes with different composition need to be further elaborated in the section, for better understanding of the science behind the same. Use suitable reference for the same.

5. The improvement in hardeness with modified ZrO2 was 29%, however, the change is TG is not of similar extent. Why? Further, with addition of PS fibre the hardness decreases, does PS lead to poor dispersability of the nanoparticles?

6. A-PS and A-PS-ZrO2/MEMO have similar hardness. Elaborate. What is the significance of using nanoparticles if only PS can improve the overall mechanical property of the matrix.

7. Indicate in the Fig 8 (mark or highlight the same in the Fig), how FE-SEM can indicate the brittle fracture tendency and its propagation.

8. A comparison table of other reported modification of the same matrix using nanoparticles with the present reported system would likely improve the quality of work.

9. The author could also cite the following articles at suitable position (for the improvement in impact strength with nanoparticle dispersions).

i. NiO nanofiller dispersed hybrid Soy epoxy anticorrosive coatings

ii. High performance anti-corrosive epoxy–titania hybrid nanocomposite coatings

iii. High-Performance Soya Polyurethane Networked Silica Hybrid Nanocomposite Coatings

6. PLOS authors have the option to publish the peer review history of their article (what does this mean?). If published, this will include your full peer review and any attached files.

Reviewer #1: No

Reviewer #2: No

---

## [Author Response · Author response to Decision Letter 0]

28 Nov 2019

Dear Editor and reviewers,

Thank you for your comments and suggestion that had helped to improve our paper.

We hope that our enclosed corrections will be full field and satisfied the requirements.

Best regards,

Autors

Reviewers' comments:

Reviewer's Responses to Questions

Comments to the Author

1. Is the manuscript technically sound, and do the data support the

conclusions?

Reviewer #1: Yes

Reviewer #2: Yes

2. Has the statistical analysis been performed appropriately and rigorously?

Reviewer #1: Yes

Reviewer #2: N/A

3. Have the authors made all data underlying the findings in their manuscript fully available?

The PLOS Data policy requires authors to make all data underlying the findings described in their manuscript fully available without restriction, with rare exception (please refer to the Data Availability Statement in the manuscript PDF file). The data should be provided as part of the manuscript or its supporting information, or deposited to a public repository. For example, in addition to summary statistics, the data points behind means, medians and variance measures should be available.

If there are restrictions on publicly sharing data—e.g. participant

privacy or use of data from a third party—those must be specified.

Reviewer #1: Yes

Reviewer #2: Yes

4. Is the manuscript presented in an intelligible fashion and written in standard English?

PLOS ONE does not copyedit accepted manuscripts, so the language in

submitted articles must be clear, correct, and unambiguous. Any

typographical or grammatical errors should be corrected at revision, so

please note any specific errors here.

Reviewer #1: No

Reviewer #2: Yes

5. Review Comments to the Author

Please use the space provided to explain your answers to the questions above. You may also include additional comments for the author, including concerns about dual publication, research ethics, or publication ethics.

Reviewer #1: This manuscript reports the preparation of a hybrid nano-composites composed of nano-zirconia and electrospum polystyrene.

Results of the studies are very interesting an

1) First three sentences of the abstract are controversial and do not present manuscript well.

Thank you for this remark. We have corrected the first sentences in Abstract: 

The processing and characterization of hybrid PMMA resin composites with nano-zirconia (ZrO2) and electrospun polystyrene (PS) polymer fibers were presented in this study. Reinforcement was selected with the intention to tune the physical and mechanical properties of the hybrid composite. Surface modification of inorganic particles was performed in order to improve the adhesion of reinforcement to the matrix.

2) Over all English needs careful proof reading

We consulted the native English lector and correction of English was done. 

3) Uniqueness aspects of the proposed work is not discussed well in introduction. 

Thank you for this remark. It was very helpful for better impact of our work. We have made reorganization of Introduction with all suggested corrections.

4) Compare the performance parameters of this system with literature.

We have done suggested comparison in Table 5:

A comparison of results presented in this paper and the literature is presented in Table 5. It is evident that for some systems of hybrid reinforcements design of mechanical properties is possible with the proper combination of constituents according to exploitation requirements. 

Table 5. Performance parameters comparison of the given system with the literature data

 Matrix Reinforcements Mechanical properties Reference

1. Acrylic Denture Base Material 1% Silanized zirconium oxide (ZrO2) nano filler and 2.5 %electrospun PS fibers

 Absorbed Impact energy improvement 70 %,

Hardness improvement 10 % This paper

2. 

 Acrylic Denture Base Material 

 Silanized zirconium oxide (ZrO2) nano filler and plasma treated polypropelene (PP) fibers tensile strength, improvement 44% 1

3. PMMA PP fibers 2.5 t.%)/Al2O3 nanoparticles (1 wt.%) Impact strength improvement 119%

Surface hardness improvement 4.2% 2

4. Denture base PMMA ZrO2-Al2O3 Tensile strength decrease 17.0% (highest value for 100% ZrO2)

Fracture toughness improvement 32.5% (ZrO2/Al2O3=20/80) 5

5. Denture base PMMA ZrO2/Aluminum borate whiskers Surface hardness improvement 26.4% (reinforcement ratio 1:2, 3% ZrO2)

Felxural strength improvement 52.3% (reinforcement ratio 1:2, 2% ZrO2) 6

6. Denture base PMMA Glass/ UHMW PE fibers in form of fabrics 

5.3% flexural strength 90% and modulus 76% 7

5) What are the prospects of this research?

Hybrid reinforcement systems have been created with aim to improve physic mechanical properties by synergy of dual reinforcements. This improvement should e better than adding them separately. So the contribution of this paper is developing of hybrid composites with designed properties according exploitation conditions. We have corrected the Introduction and added some references.

Composite materials combine the properties of their constituents offering the new material improved properties and enabling the tuning of the properties to fit predefined needs. Hybrid reinforcement composite systems are created with the aim of improving t physical and mechanical properties by a synergy of two or even more reinforcement types. In the wide area of research, hybrid reinforcements were of different combinations: particles and fibers [1- 4], two different types of particles [5], particles and whiskers [6], two or three types of fibers [7]. The improvement is, in general, better with multiple rather than a single reinforcement type, so that every one of the added reinforcements improves a different material property. One of the reinforcements should be aimed at improving toughness and the other, for example, improving hardness and elastic modulus [8]. 

The type, shape, and dispersion of fillers in a composite significantly influence the mechanical and thermal properties of the composite [9-13].

1. Ismail IJ, “Development and Performance of Composite from Modified Nano Filler with Plasma Treated Fiber and Heat Cured Acrylic Denture Base Material on Some of Its Properties – In Vitro Study“, International Journal of Science and Research (IJSR) Volume 6 Issue 3, March 2017 

2. Muklif OR, Ismail IJ, “Studying the effect of addition a composite of silanized nano-Al2O3 and plasma treated polypropylene fibers on some physical and mechanical properties of heat cured PMMA denture base material,” Journal of Baghdad College of Dentistry, vol. 27, no. 3, pp. 22–27, 2015.

3. Gad MM, Al-Thobity AM, Rahoma A., Abualsaud R, Al-Harbi AF, Akhtar S, “Reinforcement of PMMA Denture Base Material with a Mixture of ZrO2 Nanoparticles and Glass Fibers“, Hindawi, International Journal of Dentistry, Volume 2019, Article ID 2489393, 11 pages, https://doi.org/10.1155/2019/2489393

4. Gad M, Fouda S, Al-Harbi F, Napankangas R, Raustia A, “PMMA denture base material enhancement: a review of fiber, filler, and nanofiller addition,” International Journal of Nanomedicine, vol. 12, pp. 3801–3812, 2017.

5. Alhareb AO, Ahmad ZA, “Effect of Al2O3/ZrO2 reinforcement on the mechanical properties of PMMA denture base,” Journal of Reinforced Plastics and Composites, vol. 30, pp. 1–8, 2011. 

6. Zhang XY, Zhang XJ, Huang ZL, Zhu B., Chen RR, “Hybrid effects of zirconia nanoparticles with aluminum borate whiskers on mechanical properties of denture base resin PMMA,” Dental Materials Journal, vol. 33, no. 1, pp. 141–146, 2014.

7. Yu SH, Lee Y, Oh S, Cho HW, Oda Y, Bae JM, “Reinforcing effects of different fibers on denture base resin based on the fiber type, concentration, and combination,” Den. Mat. Journal, vol. 31, no. 6, pp. 1039–1046, 2012.

8. Lazouzi G, Vuksanović M, Tomić N. Z, Mitrić M, Petrović M, Radojević V, Jančić Hainemann R, Optimized preparation of alumina based fillers for tuning composite properties,Ceramics International, 7442-7449, 2018,

9. Salih SI, Oleiwi JK, Hamad QA, “Investigation of fatigue and compression strength for the PMMA reinforced by different system for denture applications,” International Journal of Biomedical Materials Research, vol. 3, no. 1, pp. 5–13, 2015.

10. Chen S, Liang W, “Effects of fillers on fiber reinforced acrylic denture base resins,” Mid-Taiwan Journal of Medicine, vol. 9, pp. 203–210, 2004.

Reviewer #2: The research work presented in the manuscript is scientifically sound and well written.

However, before accepting the same please do the following corrections:

1. Commercial acryl denture material ``Akril R`` do the needful correction.

The correction was done:

„Simgal-Acryl R“, Galenika AD, Belgrade,

2. Fe-SEM images are not very clear and do not satisfy the claimed findings. Kindly elaborate further. Also, for comparison with the effect of PS, the image 3c should also be of 500 nm scale.

Thank you for this suggestion. In Manuscript were presented samples A-PS-ZrO2; and A-PS-3% ZrO2/MEMO with 500 nm scale and only A-PS-3% ZrO2/MEMO with 5 �m scale. Now we added the photo of A-PS-ZrO2 sample with same magnification (5 �m scale). We also have corrected the omitted marks of samples. 

Fig. 3. FE-SEM images of impact fracture surface of: a) and c) A-PS-ZrO2; b) and d)) A-PS-1% ZrO2/MEMO 

3. No rationale for better dispersion of ZrO2 with surface modification and presence of PS fibre is given. Explanation required.

The followed text of FESEM description is corrected:

FE-SEM images of cross-sections of the polymer after the impact testing are presented in Fig. 3. Shows that zirconia agglomerates observed in the sample with unmodified particles had larger diameters and consisted of a larger number of individual particles, (Fig. 3a) while surface modification of nanoparticles with MEMO silane (Fig. 3b) enabled aggregates to be smaller in diameter and more evenly spaced. In Fig 3c) and 3d) the areas with fibers are presented. The modification of nano zirconia with MEMO silane produced a monolayer of silane on the surface of the particles, and promotes deagglomeration in the polymer matrix because of the steric hindrance [47].

47. Mohammadnezhad G, Dinari M, Soltani R., Bozorgmehr Z, “Thermal and mechanical properties of novel nanocomposites from modified ordered mesoporous carbon FDU-15 and poly(methyl methacrylate) “, Appl. Surf. Sci. 346 (2015) 182-188.

4. Are the DSC images plotted (not original). Provide the original DSC images. Further, how much weight of each sample was taken for the analysis. The variation in the thermal stability and changes with different composition need to be further elaborated in the section, for better understanding of the science behind the same. Use suitable reference for the same.

Thank you for this remark. We have paid attention to correct experimental part, and discussion after that. Also, we included original DSC image. Following inserted text and Literature are presented below:

Thermal analysis of composites was performed on a device for differential scanning calorimetry (DSC) in a temperature range from 24 ºC to 160 ºC (Q10, TA Instruments) under a dynamic nitrogen flow of 50 ml min–1. Samples of 7-9 mg were investigated. The samples were heated up at a rate of 10 °C min–1. The glass transition temperature was determined at the midpoint of the step-transition for each sample. The Tg values were confirmed by the use of the derivative curve.

*** 

Zirconia behave as highly functional physical cross-links, and hence reduce the overall mobility of the polymer chains, even when interactions with the polymers are only on a physical level [49, 50]. The embedding of modified nano zirconia slightly increases the Tg of the composite as a consequence of an interaction between the modified zirconia interface and acrylic resin [51]. Interfacial Si-O bond formation on the surface of zirconia enables chemical bonding with polymer matrix [50-53]. This also leads to better deagglomeration of nanoparticles. In this case, the mobility of polymer chains was suppressed even better, and Tg for this composite is the highest (Table 2). 

References:

49. Vacatello M, “Monte Carlo simulations of polymer melts filled with solid nanoparticles“, Macromolecules, 34(6) (2001) 1946–1952.

50. Thomas P, Dakshayini BS, Kushwaha HS, Vaish R, Effect of Sr2TiMnO6 fillers on mechanical, dielectric and thermal behaviour of PMMA polymer, J. Adv. Dielect. 5(2) (2015) 1550018 (11 pages) DOI: 10.1142/S2010135X15500186

51. Tommasini FJ, Cunha Ferreira L, Pimenta Tienne LG , de Oliveira Aguiar V, Prado da Silva MH, da Mota Rocha LF, de Fátima Vieira Marques M, Poly (Methyl Methacrylate)-SiC Nanocomposites Prepared Through in Situ Polymerization, Materials Research. 2018; 21(6): e20180086

52. Abboud, M, Turner M, Duguet E, Fontanille M, PMMA-based composite materials with reactive ceramic fillers Part 1.—Chemical modification and characterisation of ceramic particles. J. Mater. Chem. 7, 1527–1532 (1997).

53. Turner M, Duguet E, Labrugere C, Characterization of silane-modified ZrO2 powder surfaces. Surf. Interface Anal. 25, 917–923 (1997).

54. Otsuka T, Chujo Y, Poly(methyl methacrylate) (PMMA)-based hybrid materials with reactive zirconium oxide nanocrystals, Polymer Journal (2010) 42, 58–65

5. The improvement in hardeness with modified ZrO2 was 29%, however, the change is TG is not of similar extent. Why? 

This suggestion for discussion is very usefull for paper, thank you. We added our text in Manuscript:

Microhardness testing is based on the local plastic deformation of a sample under applied stress in the vicinity of the indenter. Glass transition temperature depends of the structure and morphology of polymer chains and its behavior in higher temperatures. In glassy state critical stress for plastic deformation of amorphous polymer requires movements of macromolecule bundles against the resistance of the stiff chain segments. , hardness for amorphous polymers could be higher than for semi-crystalline polymers at temperatures under Tg.

Further, with addition of PS fibbers the hardness decreases, does PS lead to poor dispersability of the nanoparticles? 

The PMMA and PS are immiscible and don't interact easily. Added PS fibers didn't interact with the PMMA paste during the preparation of the composite. But presence of PS fibers have influenced polymerization of PMMA, and resulted of phase separation in hybrid composite.

This leads to interfacial tension in those areas which is followed by attraction of nanoparticles in the vicinity of PS fibers. This influence of the composites’ mechanical behavior of leads to lower hardness and Tg [55-57]. 

***

Composite with surface-modified nanoparticles provides stronger resistance to plastic deformation as it is chemically bound to the matrix.

55. Chuai C, Almdal K, Lyngaae-Jørgensen J, Thermal Behavior and Properties of Polystyrene/Poly(methyl methacrylate) Blends, Journal of Applied Polymer Science, Vol. 91, 609–620 (2004)

56. Ton-That C,. Shard AG, Teare DOH, . Bradley H, XPS and AFM surface studies of solvent-cast PS/PMMA blends, Polymer, Volume 42, Issue 3, February 2001, Pages 1121-1129

57. Lee J.Y, Zhang Q, Emrick T, Crosby A.J, Nanoparticle Alignment and Repulsion during Failure of Glassy Polymer Nanocomposites, Macromolecules 2006, 39, 7392-7396

6. A-PS and A-PS-ZrO2/MEMO have similar hardness. Elaborate. What is the significance of using nanoparticles if only PS can improve the overall mechanical property of the matrix?

Hardness of A-PS was 211MPa while hardness of A-PS-1% ZrO2/MEMO was 269 MPa. Relative to pure PMMA (243 MPa) adding of PS fibers decrease hardness by 13%, while adding of nanoparticles increase by 10%.

The idea of this work was that properties of hybrid composite could be moderate by proper ratio of fibers and nanoparticles in accordance of exploitation requirements. Addition of zirconia nanoparticles should increase the hardness, while addition of nanofibers will improve toughness.

7. Indicate in the Fig 8 (mark or highlight the same in the Fig), how FE-SEM can indicate the brittle fracture tendency and its propagation.

Thank you for this remark. We added explanation of figures 8a and 8b and also for diagram at 8d which is responsible for failure modes detection. The picture from literature data is enclosed which explain the brittle to ductile modes transition.

The interfacial adhesion strength between the matrix and the fibers affected the impact property of composites and resistance to crack generation and propagation [65-69]. These failure mechanisms led to the high absorption of impact energy as a consequence of dissipation during crack propagation. FE-SEM analysis Fig. 8a, b) indicates a brittle fracture tendency in specimens with PS fibers and hybrids with untreated zirconia (river pattern associated with crack propagation).

*****

For the hybrid composites with modified zirconia, Fig. 8c, with the presence of flexible − O−Si−O− bond [24-26] consequently provide better adhesion to the matrix and cracks found difficult to propagate. Brittle to ductile transformation of failure mechanism was obtained [65-67]. 

Results of impact behavior show the possibility of composite mechanical properties modulation, Fig. 8d [70]. Incorporation of pure brittle zirconia, prone to agglomeration, reduced absorbed energy, while it was successfully compensated latter with the combination of modified particles and the electrospun polymer fibers.

70. Shah V. Handbook of Plastics Testing and Failure Analysis; John Wiley & Sons: New Jersey. 2007.

8. A comparison table of other reported modification of the same matrix using nanoparticles with the present reported system would likely improve the quality of work.

We have done suggested comparison in Table 5:

Table 5. Comparsion the performance parameters of this system with literature

 Matrix Reinforcements Mechanical properties Reference

7. Acrylic Denture Base Material 1% Silanized zirconium oxide (ZrO2) nano filler and 2.5 %electrospun PS fibers

 Absorbed Impact energy improvement 70 %,

Hardness improvement 10 % This paper

8. 

 Acrylic Denture Base Material 

 Silanized zirconium oxide (ZrO2) nano filler and plasma treated polypropelene (PP) fibers tensile strength, improvement 44% [1]

9. PMMA PP fibers 2.5 t.%)/Al2O3 nanoparticles (1 wt.%) Impact strength improvement 119%

Surface hardness improvement 4.2% [2]

10. Denture base PMMA ZrO2-Al2O3 Tensile strength decrease 17.0% (highest value for 100% ZrO2)

Fracture toughness improvement 32.5% (ZrO2/Al2O3=20/80) [5]

11. Denture base PMMA ZrO2/Aluminum borate whiskers Surface hardness improvement 26.4% (reinforcement ratio 1:2, 3% ZrO2)

Felxural strength improvement 52.3% (reinforcement ratio 1:2, 2% ZrO2) [6]

12. Denture base PMMA Glass/ UHMW PE fibers in form of fabrics 

5.3% flexural strength 90% and modulus 76% [7]

Comparison of results presented in this paper and literature is presented on Table 5. It could be seen that for some system of hybrid reinforcements design of mechanical properties is possible with proper combination of constituents according to exploitation requirements. 

9. The author could also cite the following articles at suitable position (for the improvement in impact strength with nanoparticle dispersions).

i. NiO nanofiller dispersed hybrid Soy epoxy anticorrosive coatings

ii. High performance anti-corrosive epoxy–titania hybrid nanocomposite coatings

 iii. High-Performance Soya Polyurethane Networked Silica Hybrid Nanocomposite Coatings

The suggested articles were involved in Introduction for the surface modification and discussion about the improvement in impact strength with nanoparticle dispersions: 

 Improved ceramic/polymer adhesion can be achieved with coupling agents and nano paticle coatings [18-22]. They modify the interface properties between nanoparticles and polymer matrix by attaching the modified nanoparticles to the matrix resulting in the improved mechanical and other functional properties of nanocomposites. Modification of nanoparticles could be chosen according to the polymer matrix by silanes, tetraethyl orthosilicate (TEOS), titanium isopropoxide (TIP) etc. [23-25].

****

For the hybrid composite with modified zirconia, Fig. 8c, with the presence of flexible − O−Si−O− bond [23-25] consequently have better adhesion to the matrix and cracks found difficult to propagate. Brittle to ductile transformation of failure mechanism was obtained [64-66].

---

## [Editor Report · Decision Letter 1]

3 Dec 2019

Hybrid denture acrylic composites with nanozirconia and electrospun polystyrene fibers

PONE-D-19-20656R1

Dear Dr. Tomic,

We are pleased to inform you that your manuscript has been judged scientifically suitable for publication and will be formally accepted for publication once it complies with all outstanding technical requirements.

With kind regards,

Yogendra Kumar Mishra, Ph. D.

Academic Editor

PLOS ONE
---

## [Editor Report · Acceptance letter]

10 Dec 2019

PONE-D-19-20656R1 

Hybrid denture acrylic composites with nanozirconia and electrospun polystyrene fibers 

Dear Dr. Tomić:

I am pleased to inform you that your manuscript has been deemed suitable for publication in PLOS ONE. Congratulations! Your manuscript is now with our production department. 

With kind regards,

on behalf of

Dr. Yogendra Kumar Mishra 

Academic Editor

PLOS ONE